# Thickness-Stretch Vibration of an Infinite Piezoelectric Plate with Flexoelectricity

**Yan Guo [1], Bin Huang [2],\*** and **Ji Wang [2]**

1    College of Science & Technology, Ningbo University, Ningbo 315300, China; guoyan@nbu.edu.cn
2    Piezoelectric Device Laboratory, Faculty of Mechanical Engineering & Mechanics, Ningbo University, Ningbo 315211, China; wangji@nbu.edu.cn
\*    Correspondence: huangbin@nbu.edu

**Abstract:** In this paper, the thickness-stretch vibration of an infinite piezoelectric plate is studied, with consideration of the flexoelectric effect. The theoretical model developed herein is based on a one-dimensional formulation, with the assumption that the displacement and electric potential vary through the thickness. The Gibbs energy density function and variational principle are adopted to derive the constitutive equation with flexoelectricity, governing equations, and boundary conditions. For the effect of flexoelectricity, the coupling between the strain gradient through the thickness and the electric field is considered. Two electric boundary conditions are used in this work and the corresponding frequency shift due to the flexoelectricity is calculated. The present results demonstrate that the flexoelectric effect decreases the fundamental frequency of the thickness-stretch vibration and has a significant effect on the vibrational frequencies of the thickness-stretch mode of a thin piezoelectric plate. The results also show that the flexoelectric effect has a significant size dependency, and should be taken into consideration for the design and development of next-generation high-precision and high-frequency piezoelectric transducers and resonators in the future.

**Keywords:** flexoelectricity; piezoelectricity; thickness-stretch vibration; size effect; frequency shift

## 1. Introduction

Piezoelectric materials are widely used in various functional devices due to their electromechanical coupling capabilities. They play an important role in the fields of communications, electronics, aerospace and aviation structures, and energy systems. With the rapid development of modern technology, piezoelectric devices are developing towards miniaturization and functionalization, and their requirements for material properties and structural dimensions are getting higher and higher. Nanoscale composite materials and structures will have important application potentials in future multifunctional and smart devices, due to their excellent material properties and small structure size. At the nanoscale, these materials exhibit new mechanical behaviors that are different from traditional macroscopic materials. At the nanoscale, the strain gradient of piezoelectric materials is very significant, which cannot be ignored in the design of nanoscale materials [1]. This is due to another electromechanical coupling effect, flexoelectricity, which is an electromechanical coupling effect caused by a strain gradient or non-uniform deformation and widely exists in various dielectric materials [2–4]. In fact, such non-uniformity will lead to a significant change in the macroscopic mechanical behavior of micro/nano scale piezoelectric structures with flexoelectricity.

Recent studies have found that flexoelectricity has significant impacts on the performance of micro/nano piezoelectric devices, such as Rayleigh wave propagation in semi-infinite flexoelectric dielectrics [5], Lamb wave propagation with flexoelectricity and strain gradients [6], and thickness-twist waves in nanoplates with flexoelectricity [7]. There are also many studies regarding the vibration and bending of piezoelectric nanoplates

and nanobeams which consider flexoelectricity. Shen et al. [8] investigated the converse flexoelectric effect in comb electrode piezoelectric microbeams. Yan [9] studied the size-dependent bending and vibration behaviors of piezoelectric circular nanoplates. Huang [10] investigated the flexoelectric effect on bending stresses in piezoelectric laminate under an inhomogeneous electric field.

Due to the miniaturization of high-frequency piezoelectric devices, high-frequency vibrations are also affected by flexoelectricity when the devices are thin enough [11]. As one of the main functioning modes for piezoelectric resonators and transducers, thickness-stretch mode can be adopted for film bulk acoustic resonators (FBAR) as a working mode. Two-dimensional plate equations governing the extensional and thickness-stretch coupling motions were developed by Kane and Mindlin [12]. A single equation for thickness-stretch waves can be achieved by reducing the two-dimensional plate equations of coupled extension and thickness-stretch vibration in this model. Yang [13] developed an approximate method for thickness-stretch waves in an elastic plate. He also developed an energy harvesting model based on the thickness-stretch mode [14]. Huang [15] studied the propagation of long thickness-stretch waves in piezoelectric plates by simplifying the two-dimensional coupled equations with five vibration modes. Yang et al. [16] studied the nonlinear coupling between the thickness-shear and thickness-stretch modes in a rotated Y-cut quartz resonator. Liu [17] investigated the thickness-stretch vibration of a crystal plate carrying a micro-rod array fixed on the top surface, with rods modeled by the one-dimensional structural theory for extensional vibration. In addition to these works, there are also many studies regarding nonlinear thickness-stretch vibrations [18], the thickness-stretch vibration of a magnetoelectric plate with electrodes [19], a plate under biasing field [20], the electroelastic effect [21], spurious mode analysis [22], and so on.

In this work, we present an investigation of the flexoelectric effect on the thickness-stretch vibration of an infinite piezoelectric plate. Exact solutions for pure thickness-stretch vibrations are obtained based on the one-dimensional model and two electric boundary conditions. The results reveal that the direct effect of flexoelectricity affects the thickness-stretch vibration by decreasing the fundamental frequency, and also has a significant size-dependent phenomenon. This model can aid in the understanding of the micro/nano mechanical behaviors of thickness-stretch vibrations, and can be used as guidance for the design of next-generation high-precision piezoelectric resonators.

## 2. Mathematical Modeling

### 2.1. Three-Dimensional Governing Equations

For the present work, we start with the general three-dimensional equations. For the piezoelectric plate with the consideration of flexoelectricity, we adopt the Gibbs energy density function as the energy density function for modeling. The Gibbs energy density function $H$ can be expressed as in Equation (1), which considers the coupling between the strain gradient and electric field [23].

$$H(E, S, \lambda) = -\frac{1}{2}\varepsilon_{ij}E_iE_j + \frac{1}{2}c_{ijkl}S_{ij}S_{kl} - e_{ijk}E_kS_{ij} - \mu_{ijkl}E_i\lambda_{jkl} \tag{1}$$

where $S$ is the strain tensor, $\lambda$ is the strain gradient tensor, $E$ is the electric field, $c$ is the elastic constant, $e$ is the piezoelectric constant, $\varepsilon$ is the dielectric constant, and $\mu$ is the flexoelectric coefficient.

The constitutive equations can be derived from the energy density function, as given in Equation (2).

$$T_{ij} = \frac{\partial H}{\partial S_{ij}} = c_{ijkl}S_{kl} - e_{ijk}E_k \tag{2a}$$

$$\tau_{jkl} = \frac{\partial H}{\partial \lambda_{jkl}} = -\mu_{ijkl}E_i \tag{2b}$$

$$D_i = -\frac{\partial H}{\partial E_i} = \varepsilon_{ij}E_j + e_{ijk}S_{jk} + \mu_{ijkl}\lambda_{jkl} \tag{2c}$$

where $T$ is the stress tensor, $\tau$ is the higher order stress tensor, and $D$ is the electric displacement.

The linear strains ($S_{ij}$) and strain gradients ($\lambda_{ijk}$) are calculated by the following geometric relations.

$$S_{ij} = \frac{1}{2}\left(u_{i,j} + u_{j,i}\right) \tag{3a}$$

$$\lambda_{ijk} = S_{ij,k} = \frac{1}{2}\left(u_{i,j} + u_{j,i}\right)_{,k} \tag{3b}$$

The electric field is calculated by the partial differentiation of electric potential, as given in Equation (4).

$$E_i = -\varphi_{,i} \tag{4}$$

The governing equations and boundary conditions of extended linear piezoelectricity can be derived from the variational principle [24], considering the following variational function:

$$\Pi(u,\phi) = \int_0^{t_0} dt \int_V \left[\frac{1}{2}\rho\dot{u}_i\dot{u}_i + \rho f_i u_i - H(E,S,\lambda) - \rho_e\phi\right]dV + \int_0^{t_0} dt \int_S \bar{t}_i u_i dS - \int_0^{t_0} dt \int_S \bar{\sigma}_e \varphi dS \tag{5}$$

where $\rho$, $f_i$, $\rho_e$, $\bar{t}_i$ and $\bar{\sigma}_e$ are the density, body force, volume charge density, prescribed traction force, and free charge density per unit surface area.

Calculating the first variation and the stationary condition gives the following:

$$\left(T_{ij} - \tau_{ijk,k}\right)_{,j} + \rho f_i = \rho\ddot{u}_i \text{ in V} \tag{6a}$$

$$D_{i,i} = \rho_e \qquad \text{in V} \tag{6b}$$

$$\left(T_{ij} - \tau_{ijk,k}\right)n_j = \bar{t}_i \text{ on S} \tag{6c}$$

$$D_i n_i = -\bar{\sigma}_e \quad \text{on S} \tag{6d}$$

$$\tau_{ijk}n_j n_k = 0 \quad \text{on S} \tag{6e}$$

The governing equations are reduced to the following equations by ignoring the body force and volume charge density:

$$\left(T_{ij} - \tau_{ijk,k}\right)_{,j} = \rho\ddot{u}_i \tag{7a}$$

$$D_{i,i} = 0 \tag{7b}$$

### 2.2. One-Dimensional Model for Thickness-Stretch Vibration

The three-dimensional model is reduced to a one-dimensional problem by neglecting the displacements $u_1$ and $u_2$ for the configuration shown in Figure 1. We consider the following displacement fields and electric field for the one-dimensional thickness-stretch vibration analysis of an infinite piezoelectric plate.

$$u_1 = 0, \quad u_2 = 0, \quad u_3 = u_3(x_3)\exp(i\omega t)$$
$$\varphi = \varphi(x_3)\exp(i\omega t) \tag{8}$$

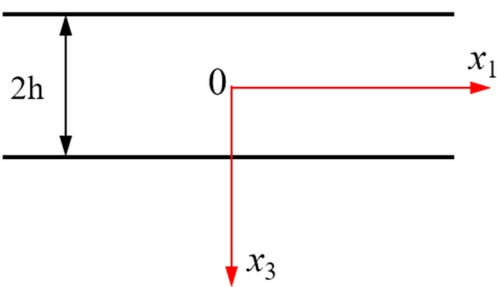

**Figure 1.** Geometry of an infinite piezoelectric plate and its coordinate system.

By means of Equations (3) and (8), the nonzero strain, strain gradient, and electric field functions are expressed below.

$$S_{33} = u_{3,3}, \ S_{33,3} = u_{33,3}, \ E_3 = -\varphi_{,3} \tag{9}$$

Additionally, the nonzero stresses and electric displacement are given below by Equation (2).

$$T_{11} = c_{13}S_{33} + e_{31}\varphi_{,3} \tag{10a}$$

$$T_{22} = c_{23}S_{33} + e_{32}\varphi_{,3} \tag{10b}$$

$$T_{33} = c_{33}S_{33} + e_{33}\varphi_{,3} \tag{10c}$$

$$D_3 = e_{33}S_{33} - \varepsilon_{33}\varphi_{,3} + \mu_{3333}S_{33,3} \tag{10d}$$

Substituting Equation (10) into Equation (5), and using the variational principle, leads to the following:

$$
\begin{aligned}
\delta\Pi(u,\phi) = {} & \int_0^{t0} dt \int_V \left[ (c_{33}u_{3,33} + e_{33}\varphi_{,33} - \mu_{3333}\varphi_{,333} - \rho\ddot{u}_3)\delta u_3 \right] dV + \int_0^{t0} dt \int_V \left[ (e_{33}u_{3,33} - \varepsilon_{33}\varphi_{,33} + \mu_{3333}u_{3,333})\delta\varphi \right] dV \\
& - \int_0^{t0} dt \iint_S \left[ \int_{-h}^h (e_{33}u_{3,3} - \varepsilon_{33}\varphi_{,3} + \mu_{3333}u_{3,33})\delta\varphi \, dx_3 \right] dS - \int_0^{t0} dt \iint_S \left[ \int_{-h}^h (c_{33}u_{3,3} + e_{33}\varphi_{,3} - \mu_{3333}\varphi_{,33})\delta u_3 \, dx_3 \right] dS \\
& - \int_0^{t0} dt \iint_S \left[ \int_{-h}^h \mu_{3333}\varphi_{,3}\delta u_{3,3} \, dx_3 \right] dS
\end{aligned}
\tag{11}
$$

Two governing equations and three boundary conditions can be obtained from Equation (11). Substituting Equation (10) into the governing equations, we can obtain the following:

$$c_{33}u_{3,33} + e_{33}\varphi_{,33} - \mu_{3333}\varphi_{,333} = \rho\ddot{u}_3 \tag{12a}$$

$$e_{33}u_{3,33} - \varepsilon_{33}\varphi_{,33} + \mu_{3333}u_{3,333} = 0 \tag{12b}$$

From the electric displacement equation of Equation (12), we can obtain the expression of $\varphi_{,33}$. Additionally, we further perform integration twice with respect to $x_3$ and differentiation with respect to $x_3$; the expressions of $\varphi$, $\varphi_{,3}$ and $\varphi_{,333}$ were obtained, and will be used for the following equations.

$$\varphi_{,33} = \frac{e_{33}u_{3,33} + \mu_{3333}u_{3,333}}{\varepsilon_{33}} \tag{13a}$$

$$\varphi = \frac{e_{33}u_3 + \mu_{3333}u_{3,3}}{\varepsilon_{33}} + B_1 x_3 + B_2 \tag{13b}$$

$$\varphi_{,3} = \frac{e_{33}u_{3,3} + \mu_{3333}u_{3,33}}{\varepsilon_{33}} + B_1 \tag{13c}$$

$$\varphi_{,333} = \frac{e_{33}u_{3,333} + \mu_{3333}u_{3,3333}}{\varepsilon_{33}} \tag{13d}$$

Substituting Equation (13) into the first equation of Equation (12), we obtain the following equation.

$$\left(c_{33} + \frac{e_{33}^2}{\varepsilon_{33}}\right) u_{3,33} - \frac{\mu_{3333}^2}{\varepsilon_{33}} u_{3,3333} = \rho \ddot{u}_3 \tag{14}$$

For the fourth-order differential equation, the general solution of Equation (14) is assumed as the following:

$$u_3 = A e^{\lambda x_3} \tag{15}$$

Substituting Equation (15) into Equation (14) results in the following characteristic equation.

$$a_1 \lambda^4 - a_2 \lambda^2 - a_3 = 0 \tag{16}$$

where

$$a_1 = \frac{\mu_{3333}^2}{\varepsilon_{33}}, \quad a_2 = c_{33} + \frac{e_{33}^2}{\varepsilon_{33}}, \quad a_3 = \omega^2 \rho$$

By defining $k = \lambda^2$, Equation (16) becomes the following:

$$a_1 k^2 - a_2 k - a_3 = 0 \tag{17}$$

The solution of Equation (17) is expressed by:

$$k = \frac{a_2 \pm \sqrt{a_2^2 + 4a_1 a_3}}{2a_1} \tag{18}$$

Thus, four solutions of the characteristic equation, Equation (16), are expressed as below.

$$\lambda_{1,2} = \pm \sqrt{\frac{a_2 + \sqrt{a_2^2 + 4a_1 a_3}}{2a_1}}, \quad \lambda_{3,4} = \pm i \sqrt{\frac{\sqrt{a_2^2 + 4a_1 a_3} - a_2}{2a_1}} \tag{19}$$

The solutions in Equation (19) are a pair of positive values and a pair of conjugate imaginary values. Thus, the general solution of Equation (14) can now be assumed as the following combination of hyperbolic and sinusoidal functions.

$$u_3 = A_1 \sinh(\alpha x_3) + A_2 \cosh(\alpha x_3) + A_3 \sin(\beta x_3) + A_4 \cos(\beta x_3) \tag{20}$$

where

$$\alpha = \sqrt{\frac{a_2 + \sqrt{a_2^2 + 4a_1 a_3}}{2a_1}}, \quad \beta = \sqrt{\frac{\sqrt{a_2^2 + 4a_1 a_3} - a_2}{2a_1}}$$

It should be noted that the frequency is contained in $a_3$, which will be solved by the following frequency functions with the help of the boundary conditions.

The next step is to obtain the frequency function from the boundary conditions. We consider two electric boundary conditions: the open circuit condition, and the short circuit condition in this work.

(a) Open circuit condition (OP)

The following traction-free boundary condition, boundary condition for higher order stress and electric displacement condition have to be satisfied for the open circuit condition:

$$T_{33} - \tau_{333,3} = 0, \quad \tau_{333} = 0 \quad (x = \pm h) \tag{21a}$$

$$D_3 = 0, \quad (x = \pm h) \tag{21b}$$

Substituting the expression of $D_3$, Equation (10d), into the third equation of Equation (21), it gives the following equation.

$$\varepsilon_{33} B_1 = 0 \tag{22}$$

The above equation implies that $B_1 = 0$ for the open circuit boundary condition. By substituting the constitutive equations, Equation (10) into Equation (21), we obtain the following four boundary equations at two surfaces.

$$\left(c_{33}\varepsilon_{33} + e_{33}^2\right)u_{3,3}(h) - \mu_{3333}^2 u_{3,333}(h) = 0 \tag{23a}$$

$$\left(c_{33}\varepsilon_{33} + e_{33}^2\right)u_{3,3}(-h) - \mu_{3333}^2 u_{3,333}(-h) = 0 \tag{23b}$$

$$e_{33}u_{3,3}(h) + \mu_{3333}u_{3,33}(h) = 0 \tag{23c}$$

$$e_{33}u_{3,3}(-h) + \mu_{3333}u_{3,33}(-h) = 0 \tag{23d}$$

Substituting the general expression of $u_3$ into Equation (23), we rearrange them in the following form.

$$\begin{aligned} & A_1\left[\alpha\cosh(\alpha h) - k_1\alpha^3\cosh(\alpha h)\right] + A_2\left[\alpha\sinh(\alpha h) - k_1\alpha^3\sinh(\alpha h)\right] \\ & + A_3\left[\beta\cos(\beta h) + k_1\beta^3\cos(\beta h)\right] - A_4\left[\beta\sin(\beta h) + k_1\beta^3\sin(\beta h)\right] = 0 \end{aligned} \tag{24a}$$

$$\begin{aligned} & A_1\left[\alpha\cosh(\alpha h) - k_1\alpha^3\cosh(\alpha h)\right] - A_2\left[\alpha\sinh(\alpha h) - k_1\alpha^3\sinh(\alpha h)\right] \\ & + A_3\left[\beta\cos(\beta h) + k_1\beta^3\cos(\beta h)\right] + A_4\left[\beta\sin(\beta h) + k_1\beta^3\sin(\beta h)\right] = 0 \end{aligned} \tag{24b}$$

$$\begin{aligned} & A_1\left[\alpha\sinh(\alpha h) + k_2\alpha^2\sinh(\alpha h)\right] + A_2\left[\alpha\cosh(\alpha h) + k_2\alpha^2\cosh(\alpha h)\right] \\ & + A_3\left[\beta\cos(\beta h) - k_2\beta^2\sin(\beta h)\right] - A_4\left[\beta\sin(\beta h) + k_2\beta^2\cos(\beta h)\right] = 0 \end{aligned} \tag{24c}$$

$$\begin{aligned} & A_1\left[\alpha\cosh(\alpha h) - k_2\alpha^2\sinh(\alpha h)\right] - A_2\left[\alpha\sinh(\alpha h) - k_2\alpha^2\cosh(\alpha h)\right] \\ & + A_3\left[\beta\cos(\beta h) + k_2\beta^2\sin(\beta h)\right] + A_4\left[\beta\sin(\beta h) - k_2\beta^2\cos(\beta h)\right] = 0 \end{aligned} \tag{24d}$$

where

$$k_1 = \frac{\mu_{3333}^2}{c_{33}\varepsilon_{33} + e_{33}^2}, \ k_2 = \frac{\mu_{3333}}{e_{33}}$$

For nontrivial solutions of the coefficients $A_1$, $A_2$, $A_3$, and $A_4$, the determinant of Equation (24) must be zero. This will lead to the frequency equation of the given problem.

(b)　Short circuit condition (SC)

For short circuit conditions, the boundary conditions are given below.

$$T_{33} - \tau_{333,3} = 0, \quad \tau_{333} = 0 \quad (x = \pm h) \tag{25a}$$

$$\varphi(x_3) = \pm\varphi_0, \quad (x_3 = \pm h) \tag{25b}$$

From the third equation of Equation (25), the potential condition leads to the following two equations.

$$\varphi(h) = \frac{e_{33}u_3(h) + \mu_{3333}u_{3,3}(h)}{\varepsilon_{33}} + B_1 h + B_2 = \varphi_0 \tag{26a}$$

$$\varphi(-h) = \frac{e_{33}u_3(-h) + \mu_{3333}u_{3,3}(-h)}{\varepsilon_{33}} - B_1 h + B_2 = -\varphi_0 \tag{26b}$$

Then, $B_1$ is obtained by the subtraction of the above two equations.

$$B_1 = \frac{\varphi_0}{h} - \frac{1}{2\varepsilon_{33}h}\left[e_{33}(u_3(h) - u_3(-h)) + \mu_{3333}(u_{3,3}(h) - u_{3,3}(-h))\right] \tag{27}$$

Similarly, the other two boundary conditions result in the following four equations at two surfaces.

$$\left( c_{33} + \frac{e_{33}^2}{\varepsilon_{33}} \right) u_{3,3}(h) + e_{33} B_1 - \frac{\mu_{3333}^2 u_{3,333}(h)}{\varepsilon_{33}} = 0 \tag{28a}$$

$$\left( c_{33} + \frac{e_{33}^2}{\varepsilon_{33}} \right) u_{3,3}(-h) + e_{33} B_1 - \frac{\mu_{3333}^2 u_{3,333}(-h)}{\varepsilon_{33}} = 0 \tag{28b}$$

$$\frac{e_{33}\mu_{3333} u_{3,3}(h) + \mu_{3333}^2 u_{3,33}(h)}{\varepsilon_{33}} + B_1 \mu_{3333} = 0 \tag{28c}$$

$$\frac{e_{33}\mu_{3333} u_{3,3}(-h) + \mu_{3333}^2 u_{3,33}(-h)}{\varepsilon_{33}} + B_1 \mu_{3333} = 0 \tag{28d}$$

Substituting the displacement function and its derivatives into Equation (28) gives the following equations, with $\varphi_0 = 0$ for free vibration.

$$\begin{aligned}
&\left[ \alpha \cosh(\alpha h) - k_3 e_{33} \sinh(\alpha h) - k_1 \alpha^3 \cosh(\alpha h) \right] A_1 \\
&+ \left[ \alpha \sinh(\alpha h) - k_3 \mu_{3333} \alpha \sinh(\alpha h) - k_1 \alpha^3 \sinh(\alpha h) \right] A_2 \\
&+ \left[ \beta \cos(\beta h) - k_3 e_{33} \sin(\beta h) + k_1 \beta^3 \cos(\beta h) \right] A_3 \\
&+ \left[ -\beta \sin(\beta h) + k_3 \mu_{3333} \beta \sin(\beta h) - k_1 \beta^3 \sin(\beta h) \right] A_4 = 0
\end{aligned} \tag{29a}$$

$$\begin{aligned}
&\left[ \alpha \cosh(\alpha h) - k_3 e_{33} \sinh(\alpha h) - k_1 \alpha^3 \cosh(\alpha h) \right] A_1 \\
&+ \left[ -\alpha \sinh(\alpha h) - k_3 \mu_{3333} \alpha \sinh(\alpha h) + k_1 \alpha^3 \sinh(\alpha h) \right] A_2 \\
&+ \left[ \beta \cos(\beta h) - k_3 e_{33} \sin(\beta h) + k_1 \beta^3 \cos(\beta h) \right] A_3 \\
&+ \left[ \beta \sin(\beta h) + k_3 \mu_{3333} \beta \sin(\beta h) + k_1 \beta^3 \sin(\beta h) \right] A_4 = 0
\end{aligned} \tag{29b}$$

$$\begin{aligned}
&\left[ \alpha \cosh(\alpha h) + k_2 \alpha^2 \sinh(\alpha h) - k_4 e_{33} \sinh(\alpha h) \right] A_1 \\
&+ \left[ \alpha \sinh(\alpha h) + k_2 \alpha^2 \cosh(\alpha h) - k_4 \mu_{3333} \alpha \sinh(\alpha h) \right] A_2 \\
&+ \left[ \beta \cos(\beta h) - k_2 \beta^2 \sin(\beta h) - k_4 e_{33} \sin(\beta h) \right] A_3 \\
&+ \left[ -\beta \sin(\beta h) - k_2 \beta^2 \cos(\beta h) + k_4 \mu_{3333} \beta \sin(\beta h) \right] A_4 = 0
\end{aligned} \tag{29c}$$

$$\begin{aligned}
&\left[ \alpha \cosh(\alpha h) - k_2 \alpha^2 \sinh(\alpha h) - k_4 e_{33} \sinh(\alpha h) \right] A_1 \\
&+ \left[ -\alpha \sinh(\alpha h) + k_2 \alpha^2 \cosh(\alpha h) - k_4 \mu_{3333} \alpha \sinh(\alpha h) \right] A_2 \\
&+ \left[ \beta \cos(\beta h) + k_2 \beta^2 \sin(\beta h) - k_4 e_{33} \sin(\beta h) \right] A_3 \\
&+ \left[ \beta \sin(\beta h) - k_2 \beta^2 \cos(\beta h) + k_4 \mu_{3333} \beta \sin(\beta h) \right] A_4 = 0
\end{aligned} \tag{29d}$$

where

$$k_3 = \frac{1}{(c_{33}\varepsilon_{33} + e_{33}^2)h}, \ k_4 = \frac{1}{e_{33}h}$$

The frequency function is obtained by letting the determinant equal zero. However, the explicit form of the frequency function is long and complicated in these formulations. Instead of obtaining the explicit form of the frequency function, we use the numerical method to search the frequencies that satisfy the conditions. We start to search the frequency from the value $\omega_0$, which is the fundamental frequency without flexoelectricity. For the open circuit condition, the resonant frequency function is given as below [19].

$$\omega_0^2 = \frac{c_{33}\varepsilon_{33} + e_{33}^2}{\varepsilon_{33}\rho} \zeta^2 \tag{30}$$

where $\zeta = \pi/(2h)$ for the fundamental frequency.

Whereas, for the short circuit condition, the resonant frequency function is given by the following equation [18].

$$\zeta h \cot(\zeta h) = \bar{k}_{33}^2 \tag{31}$$

where

$$\zeta^2 = \frac{\rho}{c_{33}(1 + k_{33}^2)} \omega_0^2, \ k_{33}^2 = \frac{e_{33}^2}{\varepsilon_{33} c_{33}}, \ \bar{k}_{33}^2 = \frac{k_{33}^2}{1 + k_{33}^2}$$

It should be noted that without flexoelectricity, the mathematical model is much simpler for analysis and more details can be found in [24].

Since the flexoelectricity affects the frequency slightly, we substitute the value of $\omega_0$ into the determinant to check whether the determinant is zero or not. The value of the frequency substituted varies from $\omega_0$ to 10% more or less than $\omega_0$, and the step must be very small to avoid missing any solutions. If the determinant is zero, the frequency substituted is the solution of the frequency functions for the given boundary conditions. By this method, we solved the frequency function instead of obtaining the explicit form.

## 3. Results

To investigate the effect of flexoelectricity on the thickness-stretch vibration of an infinite piezoelectric plate, the material properties of BaTiO$_3$ crystals were used as follows [25]: $c_{33}$ = 162 GPa, $\varepsilon_{33}$ = 12.6 $\times$ 10$^{-9}$ CV$^{-1}$m$^{-1}$, $e_{33}$ = 18.6 Cm$^{-2}$, $\rho$ = 5800 kgm$^{-3}$. The flexoelectric coefficient $\mu_{3333}$ used in their work is 10 μCm$^{-1}$. However, other papers have reported that the flexoelectric coefficients of BaTiO$_3$ can be as large as 100 μCm$^{-1}$ at a flexure frequency of 1 Hz [26]. Thus, we chose to vary the flexoelectric coefficient $\mu_{3333}$ from 10 μCm$^{-1}$ to 30 μCm$^{-1}$ in this work, to investigate the flexoelectric effect on the thickness-stretch vibration frequencies of one-dimensional micro plates.

The frequency shift of the thickness-stretch vibration with flexoelectricity was calculated in this work. The frequency shift was defined by $|\omega - \omega_0|/\omega_0 \times 10^6$ and the unit was ppm (parts per million). $\omega_0$ was the fundamental thickness-stretch mode without flexoelectricity, as given in Equations (30) and (31). Figure 2 demonstrates the results for both the open and short circuit boundary conditions. Firstly, for both conditions, the frequency was smaller than the case without flexoelectricity for the thickness-stretch vibration. This means that the flexoelectricity decreases the vibration frequency of the thickness-stretch mode. By increasing the thickness of the plate, it was found that the frequency shift decreases, implying that the flexoelectricity is size dependent and becomes much weaker as the thickness increases. For both electric boundary conditions, we observed the same tendency of linear frequency shift. The effect of the flexoelectric coefficient was also investigated by choosing different values. By increasing the flexoelectric coefficient, the frequency shift becomes large, which implies a more significant flexoelectric effect. Finally, it was observed that the frequency shift for the open circuit boundary condition was slightly more significant than for the short circuit boundary condition. Although the relative value of the frequency shift is only about 14 ppm, it is important for miniaturized high-frequency and high-precision piezoelectric devices, for the applications of next-generation transducers and resonators.

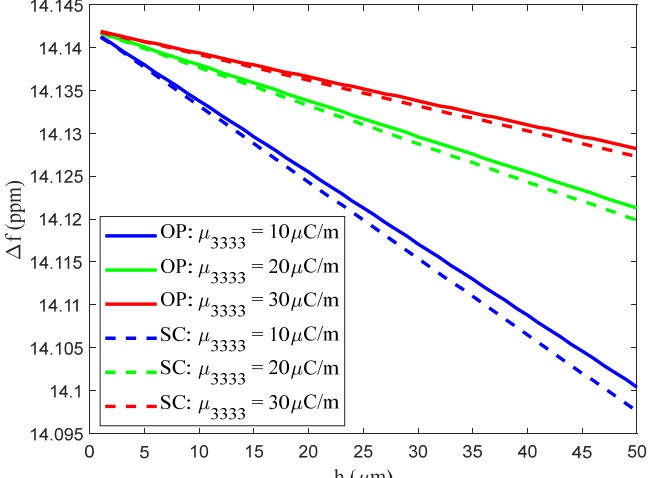

**Figure 2.** Frequency shift of fundamental thickness-stretch vibration frequency with flexoelectricity for open circuit (OP) and short circuit (SC) boundary conditions.

## 4. Conclusions

This work investigated the flexoelectric effect on the thickness-stretch vibration of an infinite piezoelectric plate. The shift of the thickness-stretch vibration frequencies of piezoelectric plates was obtained from the present model with various flexoelectric coefficients. The results show that the thickness-stretch vibration of the micro plate is affected by the flexoelectric effect, which decreases the fundamental frequency. When the thickness of the plate increases, the influence of the flexoelectric effect decreases, implying that the flexoelectric effect has a strong size dependence. The value of frequency shift is about 14 ppm for the present flexoelectric coefficients. However, this represents a significant value for the design and application of accurate high-frequency devices. The precision requirement for high-frequency piezoelectric devices can be dozens of parts per million. Therefore, we believe flexoelectricity should be taken into consideration for the design of high-precision next-generation piezoelectric devices. These results also provide important theoretical guidance and help for the design and development of accurate high-frequency piezoelectric devices in the future.

**Author Contributions:** Methodology, software, and writing—original draft preparation, Y.G.; conceptualization, and writing—review and editing, B.H.; supervision, and funding acquisition, J.W. All authors have read and agreed to the published version of the manuscript.

**Funding:** This work was funded by the National Natural Science Foundation of China (Grant no. 11702150), Natural Science Foundation of Zhejiang Province (Grant no. LY21A020003), Natural Science Foundation of Ningbo (Grant nos. 202003N4015 and 202003N4163), and the Technology Innovation 2025 Program of the Municipality of Ningbo (Grant no. 2019B10122).

**Institutional Review Board Statement:** Not applicable.

**Informed Consent Statement:** Not applicable.

**Data Availability Statement:** Not applicable.

**Conflicts of Interest:** The authors declare no conflict of interest.

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
