# Peer review of "Thickness-Stretch Vibration of an Infinite Piezoelectric Plate with Flexoelectricity"

_applsci, doi:10.3390/app12052436_

Round 1

Reviewer 1 Report

My comments for the manuscript titled:

“Thickness-stretch vibration of an infinite piezoelectric plate with flexoelectricity”

-First, English needs a full revision. There are lots of minor structural errors.

-It seems that the direct effect of flexoelectricity has been considered. However, it is not mentioned at all.

-Tensors and quantities introduced in Eqs. (1) are not suitably defined. First, by standard ε has been used for strain tensor. Moreover, cijkl is the fourth-order elasticity tensor. Other tensors shall be defined by their order and in more detail.

-“Calculating the first variation and the stationary condition is”…maybe “gives” is better than “is”

-“Gpa” should be “GPa” in all sections.

- The authors used “thickness-stretch” in the title and many parts of the paper. I think this effect shall be considered as well. Thus, in another figure, The authors can assess the plate with thickness-stretch and without it and analyze what difference this effect makes.

Author Response

We sincerely thank for his/her careful reading and valuable suggestions. The authors believe that the revised version of the manuscript reflects much better than what was intended in this work. The response can be found in the attached file.

Reviewer 2 Report

See attachment

Author Response

(The authors gave the same response as above.)

Reviewer 3 Report

In the present paper, thickness-stretch vibration of an infinite piezoelectric plate is studied.

After an introduction upon Piezoelectric materials, flexoelectric dielectrics, mechanical state of art, overview in the literature, exact solutions for pure thickness-stretch vibration are obtained based on the one-dimensional model and two electric boundary conditions.

  • The paper is well documented but begins abruptly. A description of the model, the constants involved, and the assumptions are required before defining the Gibbs energy in 2.1 as well as the velocity field in 2.2. Please give some explanations upon the boundary conditions imposed in (21).
  • An interpretation of the solution assumed in (15) that express a type of a wave, related to the determined constants and the eigenvalues in (19) should be specified before describing the final solution (20).
  • Please explain the conclusion made “The results can also provide important theoretical guidance and help for the design and development of accurate high-frequency piezoelectric devices in future.” (an idea or direction for the future).

Author Response

(The authors gave the same response as above.)
